# Differences in outcomes of mandatory motorcycle helmet legislation by country income level: A systematic review and meta-analysis

Jacob R. Lepard[1,2]*, Riccardo Spagiari[3], Jacquelyn Corley[2,4], Ernest J. Barthélemy[2,5], Eliana Kim[2,6], Rolvix Patterson[2,7], Sara Venturini[8], Megan E. H. Still[9], Yu Tung Lo[10,11], Gail Rosseau[12], Rania A. Mekary[11,13‡], Kee B. Park[2‡]

1 Department of Neurosurgery, University of Alabama at Birmingham, Birmingham, Alabama, United States of America, 2 Program in Global Surgery and Social Change, Department of Global Health and Social Medicine, Harvard Medical School, Boston, Massachusetts, United States of America, 3 Humanitas University, Milano, Italy, 4 Department of Neurosurgery, Duke University Medical Center, Durham, North Carolina, United States of America, 5 Department of Neurosurgery, Mount Sinai Health System, New York, New York, United States of America, 6 University of California-San Francisco School of Medicine, San Francisco, California, United States of America, 7 Tufts University School of Medicine, Boston, Massachusetts, United States of America, 8 Aberdeen Royal Infirmary, Aberdeen, Scotland, United Kingdom, 9 Department of Neurosurgery, University of Florida, Gainesville, Florida, United States of America, 10 Department of Neurosurgery, National Neuroscience Institute, Singapore, 11 Computational Neuroscience Outcomes Center, Department of Neurosurgery, Brigham and Women's Hospital, Harvard Medical School, Boston, Massachusetts, United States of America, 12 Department of Neurosurgery, George Washington University School of Medicine and Health Sciences, Washington, DC, United States of America, 13 School of Pharmacy, MCPHS University, Boston, Massachusetts, United States of America

‡ RAM and KBP are joint senior authors on this work.
* Jlepard@uabmc.edu

**Data Availability Statement:** All relevant data are within the manuscript and its Supporting Information files.

## Abstract

### Background

The recent Lancet Commission on Legal Determinants of Global Health argues that governance can provide the framework for achieving sustainable development goals. Even though over 90% of fatal road traffic injuries occur in low- and middle-income countries (LMICs) primarily affecting motorcyclists, the utility of helmet laws outside of high-income settings has not been well characterized. We sought to evaluate the differences in outcomes of mandatory motorcycle helmet legislation and determine whether these varied across country income levels.

### Methods and findings

A systematic review and meta-analysis were completed using the PRISMA checklist. A search for relevant articles was conducted using the PubMed, Embase, and Web of Science databases from January 1, 1990 to August 8, 2021. Studies were included if they evaluated helmet usage, mortality from motorcycle crash, or traumatic brain injury (TBI) incidence, with and without enactment of a mandatory helmet law as the intervention. The Newcastle–Ottawa Scale (NOS) was used to rate study quality and funnel plots, and Begg's and

**Funding:** This manuscript was prepared while Jacob R. Lepard, MD was a Wilson Family Clinical Scholar supported by the University of Alabama at Birmingham Women's Leadership Council. The funders had no role in study design, data collection and analysis, decision to publish, or preparation of the manuscript.

**Competing interests:** The authors have declared that no competing interests exist.

**Abbreviations:** CI, confidence interval; HIC, high-income country; IQR, interquartile range; LIC, low-income country; LMIC, low- and middle-income country; NOS, Newcastle–Ottawa Scale; OR, odds ratio; RTA, road traffic accident; SDG, Sustainable Development Goal; TBI, traumatic brain injury; WHO, World Health Organization.

Egger's tests were used to assess for small study bias. Pooled odds ratios (ORs) and their 95% confidence intervals (CIs) were stratified by high-income countries (HICs) versus LMICs using the random-effects model. Twenty-five articles were included in the final analysis encompassing a total study population of 31,949,418 people. There were 17 retrospective cohort studies, 2 prospective cohort studies, 1 case–control study, and 5 pre–post design studies. There were 16 studies from HICs and 9 from LMICs. The median NOS score was 6 with a range of 4 to 9. All studies demonstrated higher odds of helmet usage after implementation of helmet law; however, the results were statistically significantly greater in HICs (OR: 53.5; 95% CI: 28.4; 100.7) than in LMICs (OR: 4.82; 95% CI: 3.58; 6.49), $p$-value comparing both strata < 0.0001. There were significantly lower odds of motorcycle fatalities after enactment of helmet legislation (OR: 0.71; 95% CI: 0.61; 0.83) with no significant difference by income classification, $p$-value: 0.27. Odds of TBI were statistically significantly lower in HICs (OR: 0.61, 95% CI 0.54 to 0.69) than in LMICs (0.79, 95% CI 0.72 to 0.86) after enactment of law ($p$-value: 0.0001). Limitations of this study include variability in the methodologies and data sources in the studies included in the meta-analysis as well as the lack of available literature from the lowest income countries or from the African WHO region, in which helmet laws are least commonly present.

## Conclusions

In this study, we observed that mandatory helmet laws had substantial public health benefits in all income contexts, but some outcomes were diminished in LMIC settings where additional measures such as public education and law enforcement might play critical roles.

---

## Author summary

### Why was this study done?

- The utility of mandatory motorcycle helmet legislation has been studied at length in high-income country (HIC) settings, with extensive evidence demonstrating improvement in mortality and morbidity from road traffic accidents.

- Prior to this study, there was very limited discussion regarding the utility of helmet laws specifically in low-resource settings, despite the fact that over 90% of fatal road traffic injuries occurred in low- and middle-income countries (LMICs) each year.

- Based upon this, we sought to evaluate all available literature using a systematic review and meta-analysis to determine the benefit of helmet legislation in LMICs in comparison to HICs.

### What did the researchers do and find?

- A systematic review and meta-analysis were conducted, which demonstrated that road users in countries with mandatory motorcycle helmet legislation were significantly

more likely to wear a helmet and significantly less likely to experience motorcycle fatality or traumatic brain injury (TBI).

- We identified a disparity in legislative benefit, showing lower usage of motorcycle helmets and less reduction in brain injuries, including severe TBI in LMICs compared with HICs.

- Notably though, the reduction in motorcycle fatalities was similar between income contexts and overall greater in LMICs when controlling for study quality and years since law enactment, indicating that the overall goal of the law is achieved regardless of income context.

## What do these findings mean?

- Our research addresses a significant gap in the literature regarding the utility of helmet laws in low- and middle-income settings, where they are unequivocally needed the most.

- These findings indicate that helmet laws reduce mortality and have significant benefit in all income contexts.

- The presence of some disparities in legislative outcomes in LMICs highlights that additional measures beyond legislation may be needed in low-resource settings to ensure the greatest protection to the most vulnerable populations of road users worldwide.

- Our study is limited in that no studies from the lowest income countries met criteria for inclusion. This is likely a result of minimal research output or resources from these regions. Additionally, few low-income countries (13.9%) have helmet laws to be studied. We thus had to infer our findings from middle-income countries onto the lowest income countries.

## Introduction

Road traffic injuries represent a leading cause of death worldwide, taking the lives of more than 1.2 million people each year. In addition to mortality, over 50 million people on the world's roads acquire nonfatal injuries each year and are left with permanent disabilities [1]. Riders of 2-wheeled vehicles such as motorcycles are considered the most vulnerable road users along with pedestrians, as they account for more than half of all road fatalities [2]. In recent decades, the number of motorcycle users in low- and middle-income countries (LMICs) has increased dramatically, with as many as 83% to 87% of households using a motorbike as primary transportation in parts of Southeast Asia [3]. According to the World Health Organization (WHO), the majority of countries worldwide (94%) have a law in place that mandates helmet usage among motorcyclists; however, only 49 countries have comprehensive helmet laws that meet the WHO standards, which require that both drivers and passengers wear them, fastened, for all motorized 2-wheelers. The majority of these laws are present in high-income countries (HICs), with 38.4% (15/39) of HICs and only 13.9% (6/43) of low-income countries (LICs) having a comprehensive helmet law [2].

The implications of helmet legislation and their potential to reduce death and injuries related to motorcycle accidents are immense for LMICs. Over 90% of fatal road traffic injuries occur in LMICs, primarily affecting people in the working age and resulting in major economic costs to society. Mortality among motorcycle users in LMICs is more than twice than in HICs [4]. Consequently, the widespread adoption of mandatory helmet laws has been advocated by many international organizations [5,6]. While previous systematic reviews have proven the ability of helmets to prevent death and injury [7,8], little emphasis has been given to their utility in LMICs. The purpose of this study was to conduct a systematic review and meta-analysis of the literature to assess the potential utility of mandatory motorcycle helmet legislation on helmet usage, motorcyclist mortality, and incidence of traumatic brain injury (TBI) and determine if these outcomes differed across country income levels.

## Methods

### Search strategy

The study was designed and reported as per the Preferred Reporting Items for Systematic-reviews and Meta-Analysis (PRISMA) checklist (S1 Checklist) [9]. Articles were eligible for inclusion if they compared populations with and without implementation of a mandatory motorcycle helmet legislation and evaluated the association with 1 of 3 outcomes—helmet usage, motorcycle fatality, or TBI. Studies were excluded if they focused on a different vehicle type than motorcycle. A search for relevant articles was conducted on August 8, 2021 using the PubMed, Embase, and Web of Science databases. A search term was designed using appropriate key words and Medical Subject Heading (MeSH) terms to select for studies dealing with helmet legislation (S1 Table). In order to be reflective of only the most recent data, risk factors for road traffic injury, and road safety efforts, all studies published prior to 1990 were excluded. No restrictions were made regarding language of publication. Non-English articles were digitally translated into English using Google Translate (Google, Menlo Park, California, United States of America) [10].

### Study selection

After removal of duplicates, the titles and abstracts of all resultant articles were reviewed independently by 2 authors (JL and RS) for inclusion and exclusion criteria. The papers that passed this screening step were then carried forward for full-text review, which was initially divided among 5 of the authors (RS, SV, EK, MS, and RP) and then independently performed in duplicate by the primary investigator (JL). Discrepancies were addressed by discussion among reviewers to reach consensus. In the event that consensus could not be reached, all remaining discrepancies were to be resolved by the senior author (KP). There were 15 discrepancies that required additional discussion among the reviewers out of 187 articles reviewed in full text (8%, 15/187). Appropriate consensus was achieved for all 15 discrepancies without requiring further arbitration by the senior author, and 2 of these papers were ultimately included in the analysis. References for all full-text articles were reviewed to find additional relevant studies for inclusion.

### Data extraction

All articles that met criteria for inclusion underwent data extraction, which was similarly performed in duplicate, with discrepancies, if any, resolved through discussion. Information was collected from the manuscript including study characteristics including year of publication, study design, study period, country in which study and data collection occurred, income level,

when the law was enacted, enactment gap (defined as number of follow-up years post the law implementation), and specific outcomes (e.g., odds of helmet usage, odds of moderate to severe TBI, and odds of mortality related to motorcycle traffic injury). Studies were classified as including national-level data if they utilized a national database or collected data representative of the entire country, regional if they used a regional-level database or including data from multiple hospitals responsible for the care of an entire region, and single center if their data only included the clinical experience of a single hospital or institution. The World Bank Income classifications were used to assign income levels to each country studied [11]. The extracted data are available in S1 Data.

## Data analysis

The random-effects model using the DerSimonian and Laird approach [12] was used to obtain the pooled odds ratio (OR) estimates and their 95% confidence intervals (CIs) to assess the odds of helmet usage, motorcycle fatality, and TBI after helmet legislation as compared to before its implementation. Heterogeneity was statistically tested through the Cochrane Q test ($p < 0.10$) and the $I^2$ value [13,14], which estimates the percentage of variation between studies. To address potential heterogeneity sources, subgroup analysis was conducted by income level, and pooled ORs were presented for each category, along with a $p$-value that compared the pooled effect estimates across the different groups [15]. If the $p$-value was significant, the overall pooled value of all the studies was not shown; instead, only the pooled effect estimate of each group was shown. Univariate and multivariate meta-regression were used to assess whether study quality (continuous), enactment gap (continuous), and income level (binary with ref: HIC) were a significant source of heterogeneity. The "meta" and "metafor" packages in R (version 3.3.0; R Foundation for Statistical Computing, Vienna, Austria) were used to perform all analyses [16,17]. Unless otherwise specified, a $p$-value $< 0.05$ was considered statistically significant.

## Risk of bias assessment

Study quality was assessed using the Newcastle–Ottawa Scale (NOS) for observational studies assessing cohort selection, comparability, and outcome assessment [18] with a possible score range of 0 to 9. For outcomes that had at least 9 to 10 studies, the potential for small study bias was assessed through funnel plots and through Begg's and Egger's tests [19,20]. When a subgroup analysis was significant as indicated by the $p$-value, the publication bias was performed for each subgroup separately, only if the number of studies was appropriate for each subgroup.

# Results

## Search results

The initial literature search yielded a total of 3,751 documents—1,274 from PubMed, 996 from Embase, and 1,481 from Web of Science. After removing duplicates, there were 2,316 documents available to be screened by the title and abstract for inclusion. During this process, 2,127 articles were excluded due to lack of relevance to the study question, leaving 189 studies for full-text review. There were 166 articles excluded after full-text review due to the wrong intervention ($n = 69$), typically meaning that the law did not meet criteria or the intervention was not a law at all; lack of outcomes of interest being reported ($n = 49$); and noncomparative study design ($n = 48$). After searching bibliographies and references, 17 additional studies were reviewed; however, only 2 studies met criteria for inclusion in the analysis.

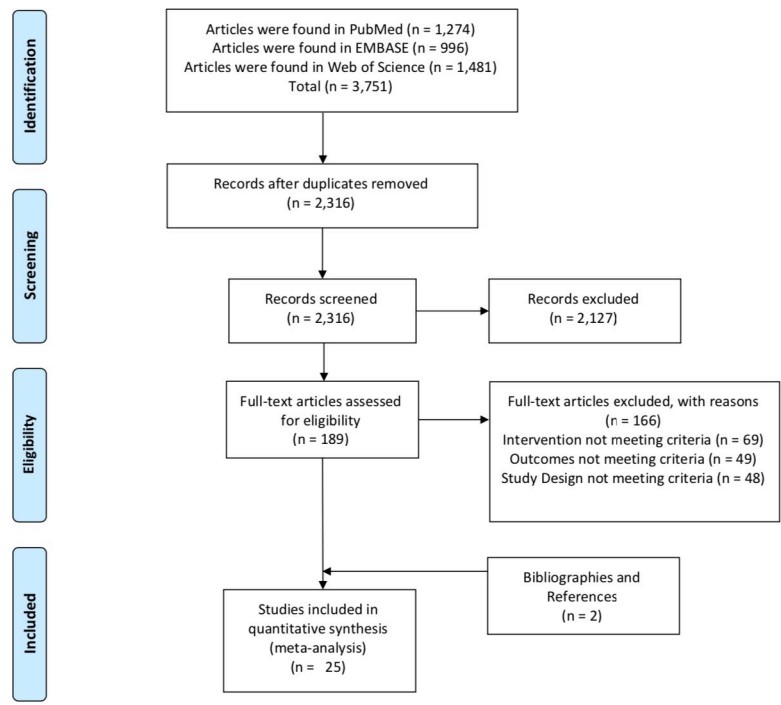

**Fig 1. Study selection process of the identified articles.** There were 17 retrospective cohort studies, 2 prospective cohort studies, 1 case–control study, and 5 pre–post design studies. One retrospective cohort study was translated from Spanish to English for data extraction [26]. All studies assessed legislation related specifically to motorcycle helmet use. Eight studies [21,23–25,28,41,42,45] utilized national-level data, another 13 studies [22,26,27,29,30,33–37,39,40,46] utilized regional-level data, and the last 4 studies [31,38,41,43] were limited to single center experience. There were 9 studies based from AMR-US/CAN, 4 studies from EUR, 1 from AMR-L, 4 from SEAR, 6 from WPR, and 1 from EMR. There were no studies evaluating laws in Africa. Sixteen of the studies evaluated helmet laws in HICs, while 4 evaluated upper-middle income, 5 evaluated lower-middle income, and none evaluated LICs. Using the NOS, the median score was 6 with a range of 4 to 9. The median quality score for studies based in HICs was 7 (IQR 5–8), while the median score for studies based in LMICs was 6 (IQR 5–6) (**Table 1**). AMR-L, Latin America; AMR-US/CAN, United States or Canada; EMR, Eastern Mediterranean region; EUR, Europe; HIC, high-income country; LIC, low-income country; LMIC, low- and middle-income country; NOS, Newcastle–Ottawa Scale; SEAR, Southeast Asia region; WPR, Western Pacific region.

Twenty-five [21–45] articles encompassing a total study population of 31,949,418 people qualified for the systematic review and meta-analysis and underwent complete data extraction (**Fig 1**). All studies utilized a comparative design either with data before and after the implementation of a helmet law in the same region (pre–post or retrospective cohort design) or simultaneous comparison between different regions with and without helmet laws (case–control or prospective cohort design). The median time interval from passage of the relevant helmet legislation to study completion (i.e., enactment gap) was 3 years with an interquartile range (IQR) of 1 to 4 years, and an overall range of 0 to 41 years. The median gap for studies based in HICs was 3 years (IQR 0.75 to 10.25), while the median gap for studies based in LMICs was 2 years (IQR 1 to 3).

## Odds of helmet usage

Ten studies with a total study population of 862,522 people evaluated the difference in motorcyclist helmet usage before and after the implementation of a mandatory helmet law. Six of these studies were based in HICs, and another 4 were based in LMICs. Four studies calculated helmet usage based upon standardized roadway observation [21,29,37,46], and another 6

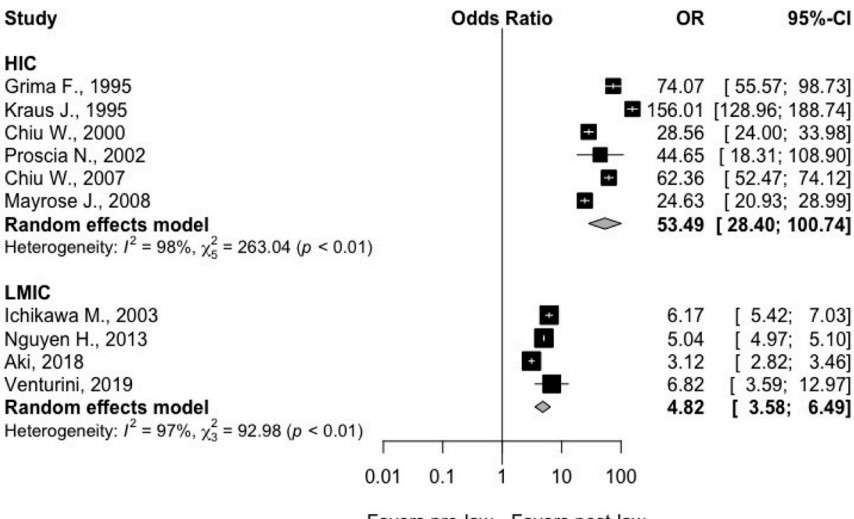

**Fig 2. Forest plots demonstrating the OR for increased helmet usage following implementation of helmet legislation (95% CI) in HICs from 6 studies and in LMICs from 4 studies.** Solid squares represent the point estimate of each study, and the centers of the clear diamonds represent the estimate of the intervention effect for HIC vs. LMIC. Horizontal lines represent 95% CIs, and the width of the diamonds represents the 95% CI of the pooled ORs. Prediction interval for the OR of helmet usage comparing post- to pre-law enactment in HIC: (5.29; 540.4). In 95% of all meta-analyses, the range of the prediction interval will capture the true effect size of 95% of all new studies in HIC. Prediction interval for the OR of helmet usage comparing post- to pre-law enactment in LMIC: (1.24; 18.7). In 95% of all meta-analyses, the range of the prediction interval will capture the true effect size of 95% of all new studies in LMIC. CI, confidence interval; HIC, high-income country; LMIC, low- and middle-income country; OR, odds ratio.

recorded helmet usage per patients hospitalized following motorcycle crash [24,25,31,34,40,43]. Although all studies demonstrated a higher odds of helmet usage after the enactment of the law, there was a stronger association upon comparing the benefit of law implementation in HICs (OR: 53.5; 95% CI: 28.4; 100.7; $I^2$: 98.0%; $p$-heterogeneity < 0.01; 6 studies) than in LMICs (OR 4.82; 95% CI: 3.58; 6.49; $I^2$: 97.0%; $p$-heterogeneity < 0.01; 4 studies), with a significant difference between the groups ($p < 0.0001$) (**Fig 2**). Although the $I^2$ was high in both categories, this could be a reflection of the change in the magnitude of the association that could differ among studies and not of the direction of the association per se. A sensitivity analysis where we removed a potential outlier (Kraus J, 1995) from the HIC subgroup did not materially alter the original results (HIC: OR: 42.5; 95% CI: 26.3, 68.5 versus LMIC: OR: 4.82, 95% CI: 3.58; 6.49).

## Odds of motorcycle fatality

Thirteen studies with a total study population of 12,830,513 reported motorcycle fatality rate as an outcome, 4 of which were based from LMIC settings. Four studies calculated fatality rate relative to the number of registered motorcycles nationally [22,23,27,28], 3 quantified based upon regional population [30,42,47], and 6 determined fatality rates per patients hospitalized following motorcycle crash [25,35,36,38,40,45]. Across all studies, there was a significantly lower odds of motorcycle fatality after enactment of the law (pooled OR 0.71, 95% CI: 0.61, 0.83; $I^2$: 68.6%; $p$-heterogeneity: 0.0001). When stratifying by income level, there was no apparent statistically significant difference comparing HICs (0.78; 95% CI: 0.66; 0.91; $I^2$: 42.0%; $p$-heterogeneity: 0.09; 9 studies) to LMICs (0.62; 95% CI: 0.44; 0.89; $I^2$: 86.2%; $p$-heterogeneity < 0.0001; 4 studies); $p$-value comparing both groups: 0.27 (**Fig 3**).

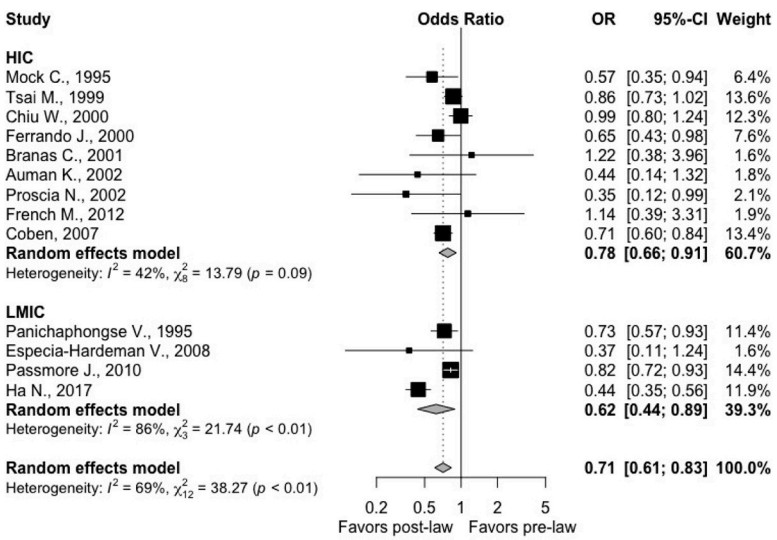

**Fig 3. Forest plots demonstrating the OR of motorcycle fatality following implementation of helmet legislation (95% CI) in HICs from 8 studies and in LMICs from 4 studies.** Solid squares represent the point estimate of each study, and the centers of the clear diamonds represent the estimate of the intervention effect for HIC vs. LMIC, while the center for the black diamond represents the overall OR for all studies. Horizontal lines represent 95% CIs for the original studies, and the width of the diamonds represents the 95% CI of the pooled ORs. Prediction interval for the OR of helmet usage comparing post- to pre-law enactment in HIC: (0.53; 1.14). In 95% of all meta-analyses, the range of the prediction interval will capture the true effect size of 95% of all new studies in HIC. Prediction interval for the OR of helmet usage comparing post- to pre-law enactment in LMIC: (0.13; 2.90). In 95% of all meta-analyses, the range of the prediction interval will capture the true effect size of 95% of all new studies in LMIC. CI, confidence interval; HIC, high-income country; LMIC, low- and middle-income country; OR, odds ratio.

## Odds of traumatic brain injury

Twelve studies with a total study population of 30,567,064 reported the incidence of TBI. When stratifying by income level, the odds of TBI after the enactment of the law was significantly more pronounced in HICs (OR: 0.61; 95% CI: 0.54; 0.69; $I^2$: 90.2%; $p$-heterogeneity < 0.0001; 10 studies) in comparison to LMICs (OR: 0.79; 95% CI: 0.72; 0.86; $I^2$: 50.8%; $p$-heterogeneity 0.15; 2 studies); $p$-value comparing both income groups was 0.0007 (**Fig 4**). Four studies [35,40,44,45] reported incidence rates specific to clinical diagnosis of severe TBI, while the other eight [24,25,32,33,36,39,41,42,48] included incidence rates broadly based on ICD-9 codes for "head injury" or "traumatic brain injury." There was no significant difference in the benefit demonstrated in those studies evaluating the odds specifically of severe TBI versus studies including all types of TBI ($p$-value: 0.32).

## Sensitivity analyses

Univariate meta-regression revealed that study quality and enactment gap were not found to have a significant effect on any of the 3 primary outcomes. Only the income level of a country was found to have a significant association with the odds of helmet usage ($p$ < 0.01) and TBI due to motorcycle accident ($p$ = 0.049), but not with the odds of motorcycle fatality ($p$ = 0.32). Notably, further adjusting for enactment gap or for study quality in the multivariate meta-regression models did not impact the statistically significant univariate results observed for income level. This adjustment did, however, impact the results observed for motorcycle fatality where the relationship with income level was found to be statistically significant ($p$ = 0.02). (**Table 2**).

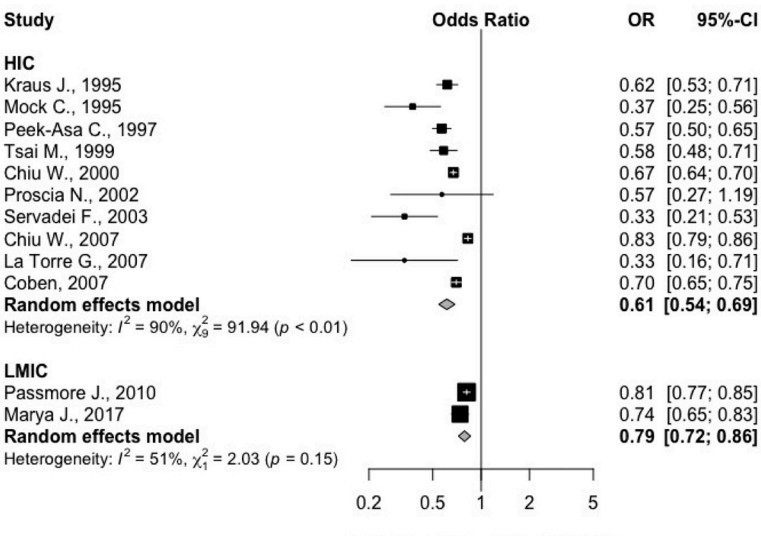

**Fig 4. Forest plots demonstrating the OR for TBI following implementation of helmet legislation (95% CI) in HICs from 9 studies and in LMICs from 2 studies.** Solid squares represent the point estimate of each study, and the centers of the clear diamonds represent the estimate of the intervention effect for HIC vs. LMIC, while the center for the black diamond represents the overall OR for all studies. Horizontal lines represent 95% CIs for the original studies, and the width of the diamonds represents the 95% CI of the pooled ORs. Prediction interval for the OR of TBI comparing post- to pre-law enactment for HIC: (0.42; 0.88). In 95% of all meta-analyses in HIC, the range of the prediction interval will capture the true effect size of 95% of all new studies in HIC. Prediction interval for the OR of TBI comparing post- to pre-law enactment for LMIC: NA due to paucity of studies. CI, confidence interval; HIC, high-income country; LMIC, low- and middle-income country; OR, odds ratio; TBI, traumatic brain injury.

## Evaluation of bias

The funnel plot was not feasible for a helmet law as each subgroup had fewer than 9 to 10 studies. For motorcycle fatality, there was no possible absence of negative studies comprising small sample sizes, as evident by the funnel plot that showed no asymmetry (**S1 Fig**) and the $p$-values of Begg (0.22) and Egger (0.29). As for the odds of TBI among studies in HICs, a slight asymmetry in the funnel plot was shown to the right side of the pooled point estimate, despite the nonstatistically significant $p$-value of Begg (0.93); however, Egger's $p$-value was significant (0.045). Notably, the source of asymmetry in a funnel plot could be due to other reasons than small study bias (e.g., true heterogeneity, data irregularities, artifactual, selection bias) [20] (**S2 Fig**).

## Discussion

In the present analysis, we found that the odds of helmet usage increased in all income contexts following passage of a helmet law, with a significantly greater benefit in HICs compared with LMICs. Studies also showed that the odds of TBI decreased by 41% in HICs, which was significantly greater than the 21% reduction in LMICs. Lastly, the odds of motorcycle fatality decreased by 29% overall with no difference among income contexts, although when controlling for study quality and enactment gap, the reduction was significantly greater in LMICs. To our knowledge, this is the first meta-analysis conducted regarding the utility of helmet legislation with specific evaluation of the benefits in LMICs compared with HICs.

In the recent Lancet Commission regarding the Legal Determinants of Health, Gostin and colleagues remark that "Law can be a powerful tool for advancing global health, yet it remains underutilized and poorly understood" [49]. According to the WHO 2018 Global Status Report

on Road Safety, there are an estimated 1.35 million deaths annually from road traffic accidents (RTAs), with a disproportionate number of these occurring in LICs. Indeed, the annual incidence of death from RTAs is 27.5 per 100,000 population in LICs compared to 8.3 per 100,000 in HICs. Since 2013, WHO has tracked metrics regarding road-related injury in an effort to meet Sustainable Development Goal (SDG) 3.6, which targets a 50% reduction in road traffic deaths by 2020 [50]. In the interval time period, 5 countries have enacted helmet legislations complying with what WHO considers to be "best practices," bringing the global total to 49 countries including 38.4% of HICs, 21.6% of MICs, and 13.9% of LICs. Despite such measures, no LICs, including those having adopted helmet legislations, have recorded a reduction in death from RTAs in contrast to 23.4% of MICs and 51% of HICs that have noted at least 2% decrease in RTA deaths since 2014 [1].

Several systematic reviews and meta-analyses have demonstrated the utility of helmets in decreasing motorcycle related deaths. A Cochrane review completed by Liu and colleagues demonstrated that helmet usage decreased the risk of death from 69% to 42% [7]. Other studies looked specifically at the potential benefits of helmet legislations as an intervention [51,52]. Du and colleagues included studies of helmet laws in multiple countries, emphasizing the global need for comprehensive laws; however, no analysis of benefit by country or income level was conducted [8]. While such analyses are important to ensure the utility of policy initiatives, the majority of studies to date have focused only on HIC contexts and thus entirely miss the global scope of the problem. The disparities of TBI prevention are clear; however, these findings beg the important question as to whether a mandatory helmet law passed in sub-Saharan Africa or Southeast Asia is an equivalent measure to a law passed in North America or Western Europe.

Our data suggest that universal passage of motorcycle helmet legislation could increase the use of helmets, reduce the incidence of TBI, and decrease motorcyclist mortality. These practical results could provide substantial progress toward achieving SDG 3.6 in a timely fashion. However, our findings of varied benefit of legislation based on the income setting suggest that there still remain context-specific barriers that prevent implementation in limited-resource settings. In particular, lack of education regarding public health interventions likely lead to poor compliance rates. In addition, the challenge of implementing such legislations with often limited law enforcement resources creates a scenario in which lawmakers may be forced to decide which policies will be given priority. This effect could explain the findings of Nazif-Muñoz and colleagues in which the benefit of road safety legislations diminished over time as law enforcement assets were reallocated to other areas [53,54]. Our data would suggest that policy makers should give the highest priority to enforcing helmet legislations, given the very high potential for preventing mortality and morbidity in young road users.

At the local level, we must first consider what impediments to compliance exist in order to provide appropriately focused incentive and education. For example, there are many cultural complexities such as religious beliefs preventing head covering [55] and misperceptions such as concern for higher risk of cervical spine injury to helmeted child passengers [56], which reduce helmet usage. Cultural norms and perceptions such as these must be addressed through culturally specific public education campaigns, which encourage individuals to be participants in their own healthcare and prevention. Such nonlegislative interventions are undeniably an important component of public health measures in all settings, but perhaps particularly in LMICs. The value of these efforts are typified by international organizations such as ThinkFirst [57] (www.ThinkFirst.org) and the Asia Injury Prevention Foundation (www.aip-foundation. org) [58], both of which have played major roles in the promotion of helmet usage and the prevention of head injury in Southeast Asia and worldwide. As helmet laws are increasingly passed in LMICs, it will be important that they are accompanied by efforts such as these.

Limited finances also represent an important practical barrier to helmet usage [59]. In general, with greater helmet quality comes higher cost and thus lower usage. While minimum standards for helmet quality and government subsidies remain useful, public education programs to teach the importance of high-quality helmet usage are an effective adjunctive strategy. In addition, motorcycle taxis have long been a fixture of roadways throughout the world and are known for unsafe driving practices that contribute to road traffic injuries [60]. Notably, as this industry is increasingly formalized via the use of app-based ride-hailing motorcycle services throughout Africa and Southeast Asia, there is a significant impetus to increase the standards of driver safety, vehicle maintenance, and helmet usage among motorcycle taxi drivers. Such trends show a significant promise in utilizing free market mechanisms to increase the usage of helmets and overall safety of commercial motorcycle drivers and passengers [61].

As are all meta-analyses, this study is limited in its reliance upon previously published literature. In order to account for any publication bias, funnel plots and trim-and-fill method were completed, when statistically feasible, for all studied outcomes and without evidence of significant bias. Additionally, among the studies included in our analyses, there were multiple study designs and data sources employed, which ranged from national-level trauma databases to individual hospital-level experiences. In an effort to provide the greatest level of transparency in the interpretation of our data, we have included in Table 1 the population scale of each study's data (single center, regional, or national) along with ratings of overall study quality using the NOS. Another limitation of the study is that all extracted data used in our analysis were summary-level data rather than individual-level data from each included study. While this has strong validity as a methodology for meta-analysis, it is possible that this could introduce bias into the data extraction process.

Notably, there was no available literature regarding helmet laws from the lowest income countries or from the African WHO region. This is likely due in part to relatively fewer LICs with helmet laws, 13.9%, but also likely represents an important regional research disparity. Given this disparity, it is possible that lower study quality and years since enactment of legislation were potential confounding variables in the results of our studies coming from LMICs. We addressed this by performing specific meta-regression analyses that demonstrated no significant effect of these variables on the 3 outcomes of interest and persistent effect of country income even when adjusting for study quality and years since law enactment. These findings suggested that study quality and years since enactment were not a significant effect modifier or source of heterogeneity in the results of our study. Moving forward, research efforts focused on the lowest-income strata should be a priority of the international community.

Our findings suggest that there are significant potential benefits to the widespread enactment of mandatory motorcycle helmet legislations in all global settings. We identified a disparity in legislative impact, showing lower usage of motorcycle helmets and less reduction in brain injuries, including severe TBI in LMICs compared with HICs. Notably though, the reduction in motorcycle fatalities was similar between income contexts and overall greater in LMICs when controlling for study quality and years since law enactment. This indicates that while some outcomes of the law are diminished in lower income settings, the overall goal of reducing mortality is achieved. The passing and enforcement of such laws provides protection to the most economically indispensable demographics through prevention of traumatic injuries. Given that LICs are, by definition, economically disadvantaged, any policy that could provide benefit to those individuals who are most capable of working a job and improving the quality and stability of a society should be given high priority. We therefore propose that not only should mandatory helmet legislation be encouraged at the international policy level but also by local champions who act as evidence-based advocates for injury prevention.

**Table 1. Summary of all included studies with study characteristics and reported outcomes with mandatory helmet legislation in effect.**

| First Author and Publication Year | Country | WHO Region | Income Level | Law Enacted | Study Period | Enactment Gap (Years) | Scale | Design | NOS |
|---|---|---|---|---|---|---|---|---|---|
| Venturini S., 2019 | Cambodia | WPR | Lower Middle | 2016[‡] | 2014–2017 | 1 | Single Center | Retro Cohort | 4 |
| Akl Z., 2018 | Lebanon | EMR | Upper Middle | 2015 | 1997–2017 | 2 | National | Pre–Post | 6 |
| Ha N., 2018 | Vietnam | WPR | Lower Middle | 2007 | 2005–2010 | 3 | Regional | Retro Cohort | 7 |
| Marya J., 2017 | India | SEAR | Lower Middle | 2014 | 2014–2015 | 1 | Regional | Retro Cohort | 6 |
| Nguyen H., 2013 | Vietnam | WPR | Lower Middle | 2007 | 2007–2011 | 4 | Regional | Pre–Post | 6 |
| French M., 2012 | USA | AMR-US/CAN | High | 1967[†] | 1988–2008 | 41 | National | Case–Control | 8 |
| Passmore J., 2010 | Vietnam | WPR | Lower Middle | 2007 | 2007–2008 | 1 | Regional | Pre–Post | 4 |
| Espitia-Hardeman V., 2008 | Colombia | AMR-L | Upper Middle | 1996, 1997 | 1993–2001 | 4 | Regional | Retro Cohort | 8 |
| Mayrose J., 2008 | USA | AMR-US/CAN | High | 1997 | 1995–2000 | 3 | Regional | Retro Cohort | 7 |
| Chiu W., 2007 | Taiwan | WPR | High | 1997 | 1991–2001 | 4 | National | Cohort | 8 |
| Coben J., 2007 | USA | AMR-US/CAN | High | 1967[†] | 2001 | 34 | National | Retro Cohort[*] | 7 |
| La Torre G., 2007 | Italy | EUR | High | 2000 | 1999–2000 | 0 | Single Center | Retro Cohort | 7 |
| Ichikawa M., 2003 | Thailand | SEAR | Upper Middle | 1994 | 1994–1997 | 3 | Single Center | Retro Cohort | 6 |
| Servadei F., 2003 | Italy | EUR | High | 2000 | 1999–2001 | 1 | National | Retro Cohort | 5 |
| Auman K., 2002 | USA | AMR-US/CAN | High | 1992 | 1990–1995 | 3 | Regional | Retro Cohort | 8 |
| Proscia N., 2002 | USA | AMR-US/CAN | High | 1967[†] | 1996–1998 | 31 | Regional | Retro Cohort[*] | 5 |
| Branas C., 2001 | USA | AMR-US/CAN | High | 1967[†] | 1994–1996 | 29 | National | Retro Cohort | 9 |
| Ferrando J., 2000 | Spain | EUR | High | 1992 | 1990–1995 | 1 | Regional | Retro Cohort | 7 |
| Chiu W., 2000 | Taiwan | WPR | High | 1997 | 1996–1998 | 3 | National | Cohort | 6 |
| Tsai M., 1999 | Taiwan | WPR | High | 1997 | 1996–1997 | 0 | National | Retro Cohort | 6 |
| Peek-Asa C., 1997 | USA | AMR-US/CAN | High | 1992 | 1991–1993 | 1 | Regional | Retro Cohort | 8 |
| Panichaphongse V., 1995 | Thailand | SEAR | Upper Middle | 1994 | 1991–1994 | 0 | Single Center | Retro Cohort | 5 |
| Mock C., 1995 | USA | AMR-US/CAN | High | 1990 | 1986–1993 | 0 | Regional | Retro Cohort[*] | 5 |
| Grima F., 1995 | Spain | EUR | High | 1992 | 1992 | 3 | Regional | Pre–Post | 5 |
| Kraus J., 1995 | USA | AMR-US/CAN | High | 1992 | 1991–1992 | 0 | Regional | Pre–Post[*] | 5 |

[‡]Cambodian helmet law passed in 2016 expanded the required population for helmet usage and increased the fines for violators in comparison to the 2009 law.

[†]The Highway Safety Act of 1966 mandated that all states in the US pass motorcycle helmet legislation. Many states have since repealed or overturned these laws allowing for comparison between these states now with no helmet law and others having had one in place for several decades.

[*]Included only severe TBI in the study.

AMR-L, Latin America; AMR-US/Can, North America–US/Canada; EMR, Eastern Mediterranean Region; EUR, Europe; NOS, Newcastle–Ottawa Scale for study quality; SEAR, Southeast Asia Region; TBI, traumatic brain injury; USA, United States of America; WPR, Western Pacific Region.

Enactment Gap = time interval from passage of the helmet legislation to study completion.

**Table 2. Meta-regression analyses of each of the 3 outcomes on each of the following trial-level covariates: enactment gap (continuous); study quality (continuous); and income level (binary with ref = HIC).**

| Outcome | Meta-regression | Slope (95% CI) | p-Value[‡] | New I² | Number of studies |
|---|---|---|---|---|---|
| Helmet usage | Enactment gap | 0.02 (−0.07, 0.11) | 0.71 | 99.7% | 10 |
| | Study quality | 0.04 (−0.66, 0.75) | 0.90 | 99.7% | |
| | Income level<br>  Ref (HIC)<br>  LMIC | _______________<br>−2.39 (−3.01, −1.77) | **<0.01** | 97.8% | |
| | Income level<br>+ enactment gap[†] | −2.41 (−3.08, −1.73)<br>−0.01 (−0.06, 0.03) | **<0.01**<br>0.61 | 98.0% | |
| | Income level + study quality[†] | −2.49 (−3.11, −1.88)<br>−0.23 (−0.52, 0.06) | **<0.01**<br>0.12 | 97.7% | |
| Motorcycle fatality | Enactment gap | 0.0001 (−0.01, 0.01) | 0.99 | 70.4% | 13 |
| | Study quality | −0.05 (−0.18, 0.09) | 0.50 | 66.8% | |
| | Income level<br>  Ref (HIC)<br>  LMIC | _______________<br>−0.18 (−0.53, 0.17) | 0.32 | 69.0% | |
| | Income level<br>+ enactment gap[†] | −0.21 (−0.61, 0.20)<br>−0.002 (−0.02, 0.01) | 0.32<br>0.72 | 68.5% | |
| | Income level + study quality[†] | −0.33 (−0.60, −0.06)<br>−0.13 (−0.24, −0.02) | **0.02**<br>**0.02** | 37.8% | |
| TBI | Enactment gap | 0.002 (−0.01, 0.01) | 0.59 | 91.1% | 12 |
| | Study quality | 0.02 (−0.06, 0.10) | 0.65 | 91.2% | |
| | Income level<br>  Ref (HIC)<br>  LMIC | _______________<br>0.23 (0.00, 0.47) | **0.049** | 89.4% | |
| | Income level<br>+ enactment gap[†] | 0.28 0.01, 0.55)<br>0.004 (−0.00, 0.01) | **0.045**<br>0.33 | 90.4% | |
| | Income level + study quality[†] | 0.35 0.08, 0.62)<br>0.07 (−0.01, 0.16) | **0.01**<br>0.09 | 85.6% | |

[‡]p-Value is obtained from the Z-test, which corresponds to the statistical significance of the slope generated by the meta-regression.

[†]Multivariate meta-regression adjusting simultaneously for income level (ref: HIC) and enactment gap (continuous). All the rest of the models are univariate meta-regression where only one trial-level covariate was entered in the model.

CI, confidence interval; HIC, high-income country; LMIC, low- to middle-income country; TBI, traumatic brain injury.

While helmet laws alone have strong potential to reduce death and disability related to RTAs, there are additional legislative reforms that could provide additional important benefit such as the creation and enforcement of speed limits, seat belt usage, and stricter policies regarding alcohol intoxication [62]. In particular, alcohol usage has been found to be a significant contributor to road-related morbidity and mortality worldwide with greater prevalence in LMIC settings [63]. Lastly, poorer road and transportation infrastructure give way to disproportionately higher rates of RTAs in LMICs [64]. Improving the ease and safety of transportation has important implications both for economics and public health and is a growing priority of the international development community. The recent Lancet Commission on Legal Determinants of Global Health makes the case that governance can provide the framework for achieving sustainable development goals and be used to implement fair and evidence-based health interventions [49]. We propose the potential public health impact that could occur with passage, and implementation of international mandatory helmet laws typifies this sentiment.

We concluded that mandatory motorcycle helmet legislation was associated with a reduction in motorcycle fatalities in all income contexts. There was improved helmet usage and

reduced TBI due to motorcycle crash worldwide, but with greater benefit seen in higher-income settings where the legal framework may be more solidified. WHO and UN have taken strong stands in better understanding the issues at hand; however, still needed are local partners that understand the data and are willing to implement policies that facilitate adequate and equitable preventive health measures to the world population.

## Supporting information

**S1 Checklist. Preferred Reporting Items for Systematic-reviews and Meta-Analysis (PRISMA) checklist.**
(DOC)

**S1 Fig. Funnel plot of standard error for log ORs of motorcycle fatality in all studies.** The vertical solid line is drawn at the pooled log OR, and the other 2 lines represent the expected 95% CI for a given standard error. The plot shows no significant publication bias. Begg's (*p*-value: 0.22) and Egger's (*p*-value: 0.29) tests showed no evidence of a statistically significant publication bias. CI, confidence interval; OR, odds ratio.
(PDF)

**S2 Fig. Funnel plot of standard error for log ORs of TBI in HICs.** The vertical solid line is drawn at the pooled log OR, and the other 2 lines represent the expected 95% CI for a given standard error. The plot shows slight asymmetry to the right of the pooled effect estimate, despite the nonstatistically significant *p*-values of Begg (0.93) and Egger (0.045). CI, confidence interval; HIC, high-income country; OR, odds ratio; TBI, traumatic brain injury.
(PDF)

**S1 Table. Search terms used for each database.**
(DOCX)

**S1 Data. Extracted data used in meta-analysis separated by study and outcome of interest.**
(XLSX)

**S1 Code. R code used for data analysis and visualization.**
(R)

## Author Contributions

**Conceptualization:** Jacob R. Lepard, Jacquelyn Corley, Ernest J. Barthélemy, Rania A. Mekary, Kee B. Park.

**Data curation:** Jacob R. Lepard, Riccardo Spagiari, Jacquelyn Corley, Eliana Kim, Rolvix Patterson, Sara Venturini, Megan E.H. Still.

**Formal analysis:** Yu Tung Lo, Rania A. Mekary.

**Investigation:** Jacob R. Lepard, Eliana Kim.

**Methodology:** Jacob R. Lepard, Jacquelyn Corley, Ernest J. Barthélemy, Rania A. Mekary, Kee B. Park.

**Supervision:** Rania A. Mekary, Kee B. Park.

**Visualization:** Rania A. Mekary.

**Writing – original draft:** Jacob R. Lepard, Riccardo Spagiari, Eliana Kim.

**Writing – review & editing:** Jacob R. Lepard, Jacquelyn Corley, Ernest J. Barthélemy, Rolvix Patterson, Sara Venturini, Megan E.H. Still, Gail Rosseau, Kee B. Park.

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
