## [Editor Report · Decision Letter 0]

11 May 2020

Dear Dr Lepard, 

Thank you for submitting your manuscript entitled "Differences in Outcomes of Mandatory Motorcycle Helmet Legislation by Country-Income Level: A Systematic Review and Meta-analysis" for consideration by PLOS Medicine.

Your manuscript has now been evaluated by the PLOS Medicine editorial staff and I am writing to let you know that we would like to send your submission out for external peer review.

Kind regards,

Artur Arikainen,

Associate Editor

PLOS Medicine

---

## [Decision Letter · Decision Letter 1]

4 Aug 2020

Dear Dr. Lepard,

Thank you very much for submitting your manuscript "Differences in Outcomes of Mandatory Motorcycle Helmet Legislation by Country-Income Level: A Systematic Review and Meta-analysis" (PMEDICINE-D-20-01817R1) for consideration at PLOS Medicine. 

Your paper was evaluated by a senior editor and discussed among all the editors here. It was also discussed with an academic editor with relevant expertise, and sent to three independent reviewers, including a statistical reviewer (reviewer #2). The reviews are appended at the bottom of this email and any accompanying reviewer attachments can be seen via the link below:

[LINK]

In light of these reviews, I am afraid that we will not be able to accept the manuscript for publication in the journal in its current form, but we would like to consider a revised version that addresses the reviewers' and editors' comments. Obviously we cannot make any decision about publication until we have seen the revised manuscript and your response, and we plan to seek re-review by one or more of the reviewers. 

We expect to receive your revised manuscript by Aug 25 2020 11:59PM. Please email us (plosmedicine@plos.org) if you have any questions or concerns.

We look forward to receiving your revised manuscript. 

Sincerely,

Emma Veitch, PhD

PLOS Medicine

On behalf of Clare Stone, PhD, Acting Chief Editor,

PLOS Medicine

plosmedicine.org

*The academic editor advising us noted the difference of opinion between reviewers and commented "Reviewer #2 and #3 are scholarly, constructive, yet harsh.. my judgments align more closely to Reviewer #1 and I believe the article might be salvaged after major revisions based on Reviewer #2 and #3 recommendations".

*Please structure the abstract using the PLOS Medicine headings (Background, Methods and Findings, Conclusions) - "Methods and Findings" is a single subsection. 

*In the last sentence of the Abstract Methods and Findings section, please include a brief note about any key limitation(s) of the study's methodology.

*As noted by reviewers, PRISMA is a reporting tool rather than a guideline for *conduct* of systematic reviews. In the abstract and Methods section, it's implied that PRISMA was used to design the study rather than to guide reporting - this could be rephrased.

Comments from the reviewers:

Reviewer #1: Please see attached my comments for authors

Reviewer #2: See attachment

Michael Dewey

Reviewer #3: There is a concern that road safety legislation could have differential impact for high vs low/middle income countries. So, the topic of this systematic review and meta-analysis is a good one. However, I have tried to make sense of the data extracted and the analysis performed, but I have not had much luck. 

There are at least two ways this data could be analysed: (1) assume the log OR is normally distributed which would also require extracting its variance or both can be computed if the data are available, or (2) random effects logistic regression which are possible if the study-level counts are available (they appear to be). I am assuming the authors did approach (1) since there is mention of the DerSimonian and Laird estimator for the random effect tau^2. But, in doing so, there is no natural interaction term to be analysed if the authors wanted to, say, compare high vs low/middle income countries. In a logistic regression model, it would be possible to have main effects for helmet law (y/n) and HIC (y/n) and their interaction term. It is possible to add HIC (y/n) as a moderator to approach (1) but this is not explicitly an interaction term. So, I am left with not fully understanding what the authors did for the analysis. My preference would be approach (2) as this would likely be the analysis for each study given a binary outcome and the desire to adjust for other factors.

In the supplementary data file, counts are given for helmet use and other variables before and after motorcycle helmet legislation with, I assume, N representing the population or total sampled and n representing "cases". This notation becomes confusing when representing rates. For example, the Tsai (1999) fatality rate is 307/11000000 (per population? or per motorcyclist?) from the data, I think. This is problematic if using approach (1) from above. The variance for the log OR (or even the log odds for each group) requires the actual count and not the rate to compute the variance. It could be argued that 1/(11000000-307) (the contribution to the log OR variance) is a very small number and would be similar if the full count were known, but this should be detailed/argued somewhere in the submission.

The categorisation of countries is unclear. For example, India does not belong in Southeast Asia (SEAR) or Western Pacific Region (WPR), although listed as SEAR in the data. Vietnam is listed separately as being in SEAR and WPR, depending on the study. The US is listed as "United States" and "USA". These inaccuracies could influence the analysis if not corrected (modelling software would treat them as different countries). Why not just list out the 10 countries?

In my opinion, the conclusions of this review are unsupported until the data are corrected and the analysis performed is made clear. I would also urge the authors to provide code used to perform the analyses. The authors mention using CMA to perform analyses, and I would recommend using freely available software like R and the meta-analysis package metafor. This would make it easier to check if the analyses were performed correctly.

Other issues:

Abstract, Background: Why and how have helmet laws been undermined in LMIC's? That is a bold statement that should be supported by evidence.

Abstract, Methods: Please use "crash" instead of "accident". It is unlikely known whether a crash was intentional or not.

It is not clear what "p-interaction" or "p-heterogeneity" mean? I can certainly guess at it, but these are not standard terms for p-values associated with tests of interaction or heterogeneity, and I strongly urge they are not adopted. It only creates confusion. 

Also, see my point above, there is no clear interaction term in a meta-analysis model for the log OR with HIC (y/n) as a moderator. Assuming there is an interaction term being tested, there are better ways of assessing whether it should be included in a model or not (e.g., likelihood ratio test, Akaike or Bayesian information criterion). 

Abstract, Conclusion: The first sentence is background info and should be given above it.

Assuming the data and the analyses are correct, then the smaller impact of motorcycle TBI and fatalities in LMIC's are expected due to less uptake of helmets. This should be part of the conclusions.

Introduction. No citation(s) given for the first two sentences. Where do these statistics come from? It is not common knowledge that 1.2M people die on roads each year. There is a ref #1 that does not appear to have been used.

No quotes are needed for vulnerable road users.

What percentage of HIC's have comprehensive helmet laws?

This study is an assessment of the effectiveness of motorcycle helmet laws. Efficacy, in this context, would be whether a helmet was effective in biomechanical studies (e.g., dummy drop test).

PRISMA is a checklist not a guideline, although it is sometimes used that way (note: PRISMA checklist not provided in supplementary material).

How did the authors decide the start date for the search? Why not 1980? Or 2000? There do not appear to be many studies on this topic, so it is not clear why a start date was used at all.

It should made clear in the methods what the plan is for non-English studies. It shows up in the results, but presumably there was a plan before studies were identified (who translated the Spanish article?).

Data extraction: What is "country of focus" mean? Presumably the authors mean country of data collection?

Data analysis: 

The trim-and-fill method should never be used to "correct" for publication bias. 

https://pubmed.ncbi.nlm.nih.gov/26186117/

https://pubmed.ncbi.nlm.nih.gov/12820277/

http://www.metafor-project.org/doku.php/plots:funnel_plot_with_trim_and_fill

Be careful with terminology. The search would yield abstracts and other documents that are not "articles" and documents are "screened" by the title and abstract for inclusion. 

Results, 2nd paragraph. Presumably 24 articles qualified for inclusion and then data were extracted? What is a "single centre experience"?

Authors state in results: "All studies assessed legislation related specifically to motorcycle helmet use." Shouldn't this have been an inclusion criterion? It would be odd if this were not the case.

I urge the authors be cautious in over-interpreting "tests for funnel plot asymmetry" as "tests for publication bias". There is a desire to assume these are the same, but they are not. A funnel plot can be asymmetric for reasons other than publication bias, e.g., unaccounted for heterogeneity.

[LINK]

---

## [Decision Letter · Decision Letter 2]

16 Oct 2020

Dear Dr. Lepard,

Thank you very much for submitting your manuscript "Differences in Outcomes of Mandatory Motorcycle Helmet Legislation by Country-Income Level: A Systematic Review and Meta-analysis" (PMEDICINE-D-20-01817R2) for consideration at PLOS Medicine. 

Your paper was evaluated once more by a senior editor and discussed among all the editors here. It was also discussed with an academic editor with relevant expertise, and sent to the three independent reviewers who reviewed previously, including a statistical reviewer. The reviews are appended at the bottom of this email and any accompanying reviewer attachments can be seen via the link below:

[LINK]

In light of these reviews, I am afraid that we will not be able to accept the manuscript for publication in the journal in its current form, but we would like to consider a revised version that addresses the reviewers' and editors' comments. Obviously we cannot make any decision about publication until we have seen the revised manuscript and your response, and we plan to seek re-review by one or more of the reviewers. 

We expect to receive your revised manuscript by Nov 06 2020 11:59PM. Please email us (plosmedicine@plos.org) if you have any questions or concerns.

We look forward to receiving your revised manuscript. 

Sincerely,

Artur Arikainen

Associate Editor 

PLOS Medicine

plosmedicine.org

1. Please respond to reviewer #3’s comments below. Specifically, we would ask that you provide the underlying data, calculations, and code in a format amenable to review. Further consideration of your manuscript will likely depend on satisfactory resolution of these concerns.

2. Abstract:

a. Please report your abstract according to PRISMA for abstracts, following the PLOS Medicine abstract structure (Background, Methods and Findings, Conclusions) http://www.plosmedicine.org/article/info:doi/10.1371/journal.pmed.1001419 .

b. Please report the databases searched, the search date ranges, inclusion criteria, scales used for quality/bias assessment.

c. Please report the overall quality of included studies, and a break-down by region and study design.

d. In the last sentence of the Abstract Methods and Findings section, please describe the main limitation(s) of the study's methodology.

3. At this stage, we ask that you include a short, non-technical Author Summary of your research to make findings accessible to a wide audience that includes both scientists and non-scientists. The Author Summary should immediately follow the Abstract in your revised manuscript – you may repurpose your “Research in Context” section for this, but please follow our format. This text will be subject to editorial change and should be distinct from the scientific abstract. Please see our author guidelines for more information: https://journals.plos.org/plosmedicine/s/revising-your-manuscript#loc-author-summary

4. Line 230: PLOS does not permit data "not shown”; please remove this claim, or do one of the following:

a) If you are the owner of the data relevant to this claim, please provide the data in accordance with the PLOS data policy, and update your Data Availability Statement as needed.

b) If the data not shown refer to a study from another group that has not been published, please cite personal communication in your manuscript text (it should not be included in the reference section). Please provide the name of the individual, the affiliation, and date of communication. The individual must provide PLOS Medicine written permission to be named for this purpose.

c) For any other circumstance, please contact the journal office ASAP.

5. Thank you for providing your PRISMA checklist. Please replace the page numbers with paragraph numbers per section (e.g. "Methods, paragraph 1"), since the page numbers of the final published paper may be different from the page numbers in the current manuscript. Please rename the file ‘S1 Checklist’ and cite it in the Methods, eg.: "This study is reported as per the Preferred Reporting Items for Systematic Reviews and Meta-Analyses (PRISMA) guideline (S1 Checklist)."

----

Comments from the reviewers:

Reviewer #1: Dear authors,

I am extremely pleased by your professionalism when answering all my requests. I look forward to read this very important manuscript.

Reviewer #2: The authors have addressed my points

Michael Dewey

Reviewer #3: I do not believe the authors have properly addressed my initial concerns. So, I would like to first restate my main concern:

* In my opinion, the conclusions of this review are unsupported until the data are corrected and the analysis performed is made clear.

My suggestion was to reproduce the analysis using open source software such as R. Having done so, with results at least similar to those using CMA, I would have been satisfied and that would be the end of it for me. I cannot check the authors' CMA code (assuming there is any and not just point-and-click software). That is, this is an issue of reproducibility and not about software choices.

Instead of addressing this issue, the authors have decided to make false claims about who develops statistical software. With all due respect to Larry Hedges and Julian Higgins, the primary developers of R are truly world-renowned statisticians, from initial developers Ross Ihaka and Robert Gentleman (Department of Statistics, University of Auckland) to current developers like Hadley Wickham (last year's COPSS Presidents' Award winner; sometimes called the "Nobel Prize of Statistics"). Also note that I am the head of the statistics department at my university, and our statistics courses are primarily taught using R.

But "whose statisticians are the best" was not my criticism of the submission. I found discrepancies in the data and the authors' response has done little to alleviate those concerns. I do appreciate the column "Methodology Comments (Denominator)" that has been added to the data, but it also raises concerns the effect sizes are heterogeneous because of the denominators used. From what I can gather, there are at least five versions of this total -- population, motorcycle registrations, fatal crashes, reported injuries, and admitted patients. The odds (and therefore odds ratio) are not equivalent when comparing the numbers of cases to each of these. This would be further problematic if the choice of N were confounded with LMIC (yes vs no). The authors should try accounting for these differences, if possible, and include this as a limitation at a minimum.

Re term "p-interaction": It is somewhat clear that comparisons are between helmet law and HIC, but this is not a formal test for interaction. The use of the term "p-interaction" is not justified and very confusing, especially when no model interaction term is being assessed. At a minimum, this non-standard terminology needs to be defined including in the abstract since it appears there. Also note that the Borenstein & Higgins paper cited by the authors makes no mention of "p-interaction" or even "interaction" at all in their paper. I think the terms "effect modifier" and "source of heterogeneity", as used by the authors in their response, is much easier to understand and would be considered standard terminology. 

I do think it is acceptable to perform a meta-analysis using summary data, especially when the counts are unavailable as is the case here. However, the authors need to make this clear in the text and it should be listed as a limitation.

I believe the authors should have considered a multivariate meta-regression model as there are likely effect sizes that are not independent. I do not use CMA, but I could not find where this is a feature on their website or their manual.

https://www.meta-analysis.com/pages/features.php?cart=BGM44889920

https://www.meta-analysis.com/downloads/MRManual.pdf

Response to Other Issues #4: A meta-regression can be fit by maximum likelihood, so likelihood methods (e.g., likelihood ratio test, AIC) are possible. It is not true that the width of the confidence interval is a valid measure to assess the model goodness of fit. The summary log OR (not untransformed OR as in response) is computed as a weighted average for both fixed effects and random effects meta-analysis. The "optimal" weights among unbiased estimators (that is, smallest variance) can be shown to be those using the inverse variance (with/without tau^2, depending on approach). This was proven many years ago by Larry Hedges. So, the resulting summary estimator will have the smallest variance and therefore narrowest confidence interval. That is, the CI width is the smallest by construction and not because of how well a model fits the data.

---

[LINK]

---

## [Decision Letter · Decision Letter 3]

7 Jan 2021

Dear Dr. Lepard,

Thank you very much for submitting your revised manuscript "Differences in Outcomes of Mandatory Motorcycle Helmet Legislation by Country-Income Level: A Systematic Review and Meta-analysis" (PMEDICINE-D-20-01817R3) for consideration at PLOS Medicine. 

[LINK]

In light of these reviews, we would like to once more consider a revised version that addresses the reviewers' and editors' comments. Obviously we cannot make any decision about publication until we have seen the revised manuscript and your response, and we plan to seek re-review by one or more of the reviewers. 

We expect to receive your revised manuscript by Jan 28 2021 11:59PM. Please email us (plosmedicine@plos.org) if you have any questions or concerns.

We look forward to receiving your revised manuscript. 

Sincerely,

Artur Arikainen, 

Associate Editor 

PLOS Medicine

plosmedicine.org

1. We note that one reviewer still has some concerns regarding your results. Please therefore provide the latest version of the underlying data, to allow for verification. Please name the file(s) S1 Data etc, and provide a legend at the end of the manuscript.

2. Abstract: 

a. Please combine the Methods and Findings sections into one section, “Methods and findings”.

b. Here and throughout, please give exact p values above 0.001, and P<0.001 otherwise.

c. Please include another limitation at line 31.

d. Please delete the Key Words section at lines 36-37.

3. Please provide supplementary tables without tracked changes, and ensure all have a unique number starting from 1, along with a legend at the end of the manuscript.

Comments from the reviewers:

Reviewer #3: Thanks for the detailed response and I appreciate the inclusion of a statistical author. However, the results from CMA and R differ which should not be the case. The DerSimonian-Laird estimator is being used to estimate the random effect (as indicated in R code). This estimator has a closed form solution, so there should be no discrepancies between R and CMA (i.e., no differences in rounding or optimisation procedure). This could be due to using DL when using R and perhaps using ML or ReML when using CMA, but this is not clear.

I attempted to investigate, but the data used has not been provided. I downloaded the data in an earlier submission which I understand has been corrected. The latest version of the data, as far as I can tell, is not part of the submission. So, I cannot perform my review until this is addressed. By the way, uploading the data and the R code file would be helpful and expedite the process.

I am not purposely trying to be obstinate, but I have had serious concerns about how the meta-analysis was conducted from the beginning and those concerns have not been allayed through any of the revisions.

[LINK]

---

## [Decision Letter · Decision Letter 4]

18 Mar 2021

Dear Dr. Lepard,

Thank you very much for re-submitting your manuscript "Differences in Outcomes of Mandatory Motorcycle Helmet Legislation by Country-Income Level: A Systematic Review and Meta-analysis" (PMEDICINE-D-20-01817R4) for review by PLOS Medicine.

I have discussed the paper with my colleagues and the academic editor and it was also seen again by the 3rd reviewer. We can't proceed unless the points below are dealt 

[LINK]

We look forward to receiving the revised manuscript by Mar 25 2021 11:59PM.   

Sincerely,

Dr Raffaella Bosurgi

Executive Editor 

PLOS Medicine

plosmedicine.org

Requests from Editors:

Comments from Reviewers:

Reviewer #3: Thanks for going through this process again and putting a lot effort into my queries.

I have played around with the data the authors provided and the meta-analysis functions in R meta and metafor. 

First, I think it is overly simplistic to claim results differ across software packages for, presumably, computational differences. Generalized linear models do not have closed form solutions, but running some sample code for logistic regression across R and SAS give the exact same results to 4 decimal places. A few notes about this. I set the seed in R to randomly generate data which I then imported into SAS (i.e., data files are the same). The differences are not in the estimates but in the number of significant digits each program prints by default. 

My code:

R:

set.seed(12345)

y = round(runif(100),digits=0)

x = rnorm(100)

dta <- data.frame(y=y,x=x)

write.csv(dta,'test_data.csv',row.names=F)

reg <- glm(y~x,family='binomial',data=dta)

summary(reg)

SAS:

proc import file='test_data.csv'

 out=test

 dbms=csv

 replace;

run;

proc logistic data=test;

class y;

model y = x;

run;

This is also true for meta and metafor for a simple case. Using the authors' data with no moderator for country's income, the DL approach provides the exact same results (estimates for tau^2 and its se below).

> c(res.meta$tau2,res.meta$se.tau2)

[1] 1.423870 1.238741

> c(res.metafor$tau2,res.metafor$se.tau2)

[1] 1.423870 1.238741

This is not the case when moderators or subgroups are included. The overall estimates are similar, but this is not a computational issue. I could find much about how each package estimates tau^2 with subgroups, but the metagen() function in meta gives this message:

Details on meta-analytical method:

- Inverse variance method

- Maximum-likelihood estimator for tau^2

- Q-profile method for confidence interval of tau^2 and tau

The methods used in metafor are less clear but they are different from the meta package. Since CMA is propriety, it is less clear what methods they use.

Note that I also tried using maximum likelihood for the two packages and got similar but not exactly the same results.

In sum, I believe the available software -- CMA, R meta, R metafor -- are fitting different models to a DL-type meta-analysis model with a single moderator. It is not clear to me which is the best or better of them, but in some sense they could all be reasonably argued to be acceptable. But this does mean the available software is making different modelling assumptions which may or may not be acceptable for this data.

It is important that scientific studies be transparent and the results be reproducible. Therefore, my recommendation to the authors is to reproduce all their analyses using their data using either R meta or R metafor, and not use CMA. They should also make their data set and R code for ALL analyses publicly available. This will make the analytic approach as transparent as possible while confirming the results can be reasonably reproduced from the data on hand. 

In making this recommendation, I would to make it clear that I find it a bit disturbing software packages, especially proprietary ones, do no make it clear what models are being fit and how things are computed. Also, this issue is of no fault to the authors of the submission, so I urge them to be as transparent as possible to avoid any later confusion.

[LINK]

---

## [Decision Letter · Decision Letter 5]

6 Aug 2021

Dear Dr. Lepard,

Thank you very much for re-submitting your manuscript "Differences in Outcomes of Mandatory Motorcycle Helmet Legislation by Country-Income Level: A Systematic Review and Meta-analysis" (PMEDICINE-D-20-01817R5) for review by PLOS Medicine.

I have discussed the paper with my colleagues and the academic editor and it was also seen again by two reviewers. I am very pleased to say that provided the remaining editorial and production issues are dealt with we are planning to accept the paper for publication in the journal.

[LINK]

We look forward to receiving the revised manuscript by Aug 13 2021 11:59PM.   

Sincerely,

Louise Gaynor-Brook, MBBS PhD

Associate Editor 

PLOS Medicine

plosmedicine.org

Requests from Editors:

General comments:

Please avoid using ‘effect’ and ‘effectiveness’ throughout your manuscript, as these should be used only when causality can be inferred, i.e. from an RCT. 

Abstract:

Please report your abstract according to PRISMA for abstracts, following the PLOS Medicine abstract structure (Background, Methods and Findings, Conclusions) http://www.plosmedicine.org/article/info:doi/10.1371/journal.pmed.1001419

Abstract Background:

Line 8 - please replace ‘effects of’ with ‘differences in outcomes of’. 

Line 9 - please replace ‘ the effectiveness’ with ‘these’ to avoid using effect/effectiveness

Abstract Methods and Findings:

Please include details on the participant numbers of the studies included if possible; preferably also for the main results presented. 

We require that SRs are updated to within roughly 6 months of the expected publication date. Please update your search to the present time.

Line 22 - For the OR of helmet usage, please specify the comparison group. Please revise ‘ effect’

Abstract Conclusions:

Please begin your Abstract Conclusions with "In this study, we observed ..." or similar, to summarize the main findings from your study.

Line 33 - please clarify what is meant by ‘ typified this sentiment’ 

Author Summary:

Line 54 - please revise ‘effectiveness’

In the final bullet point of ‘What Do These Findings Mean?’, please describe the main limitations of the study in non-technical language.

Introduction:

Line 91 - please expand upon ‘its effectiveness for... '

Line 93 - please consider another term for ‘inflicting’ 

If there has been a systematic review of the evidence related to your study, please refer to and reference that review and indicate whether it supports the need for your study 

Methods:

We require that SRs are updated to within roughly 6 months of the expected publication date. Please update your search to the present time.

For those less familiar with the Newcastle-Ottawa Scale, please provide the maximum score possible 

When completing the PRISMA checklist, please use section and paragraph numbers, rather than page numbers which may not correspond to the appropriate sections after copy-editing.

Results

Please include details on the participant numbers of the studies included if possible; preferably also for the main results presented. 

Discussion:

Please re-organize the Discussion so that implications and next steps for research, clinical practice, and/or public policy follow the strengths and limitations of the study; followed finally by a one-paragraph conclusion.

Please remove all subheadings within your Discussion 

Lines 257, 294-6 - please consider another term for ‘dramatically’ 

Line 334 - please temper the assertion that your findings were ‘unequivocal’

Figures:

Figure 2 - as pointed out by the reviewer, please do check the labels on the x axis. Please also check that the prediction interval for OR in HIC has been quoted correctly: 5.29 - 540.4

Tables:

Table 1 - please clarify which study only included TBI (should be marked with * )

Table 2 - When a p value is given, please specify the statistical test used to determine it.

Comments from Reviewers:

Reviewer #2: I have reviewed this before but I think I skipped a round of commenting so I am just evaluating the current submission. I have no problems with the current version and I think the degree of detail provided should enable anyone who disagrees with the analysis to replicate it or carry out something different.

I am not so concerned with the problems of differences between software of the estimation of τ² even when using the same method as I believe any differences would be less than using a completely different estimation method. These can give surprisingly different results. However it may be interesting to visit a thread on the R-sig-meta-analysis mailing list which discusses what may be a relevant difference between meta and metafor. However it does date from 2017 and both packages are in constant development.

Michael Dewey

Reviewer #3: I am confused by parts of the authors' response. The authors state:

"we have provided our original data and R-code for all analyses performed"

However, the file "Helmet-R-Script_V4.docx" contains one analysis/forest plot/check for publication bias. I believe there should be three sets of analyses in this file. Additionally, it is unclear why R code has been saved as a WORD file. Why not save it in its native format?

Figure 2: The text states: "all studies demonstrated a higher odds of helmet usage after the enactment of the law"; however, Figure 2 indicates, effectively, a decrease in helmet usage post-law (all OR's are within the "Favors pre-law" region). Is something wrong with your code or the labelling of the figure?

Given the current forest plots in the submission, this statement in the author summary does not really appear true: "Interestingly, the data demonstrated significantly lower effectiveness in LMIC settings". The submission does support a smaller increase in helmet wearing (if Fig 2 was mislabeled, OR=4.82 vs OR=53.49) and less benefit from TBI but STILL a benefit (Fig 4, OR=0.79 vs 0.59), but LMIC's have larger reduction in fatalities than HIC's (OR=0.62 vs 0.79). That is, the results are a mixed bag with regards to LMIC's vs HIC's. I am sure many would argue larger reductions in fatalities is the primary goal of many road safety interventions like helmet laws.

Please make sure all errors have been corrected in this paper, so that we can all move on from this.

[LINK]

---

## [Editor Report · Decision Letter 6]

3 Sep 2021

Dear Dr Lepard, 

On behalf of my colleagues and the Academic Editor, Prof. Donald Redelmeier, I am very pleased to inform you that we have agreed to publish your manuscript "Differences in Outcomes of Mandatory Motorcycle Helmet Legislation by Country-Income Level: A Systematic Review and Meta-analysis" (PMEDICINE-D-20-01817R6) in PLOS Medicine.

PRESS

Sincerely, 

Louise Gaynor-Brook, MBBS PhD 

Associate Editor 

PLOS Medicine